# Extrapolation by Association: Length Generalization Transfer in Transformers

**Ziyang Cai**[*]
University of Wisconsin-Madison

**Nayoung Lee**
University of Wisconsin-Madison

**Avi Schwarzschild**
Carnegie Mellon University

**Samet Oymak**
University of Michigan

**Dimitris Papailiopoulos**
University of Wisconsin-Madison
Microsoft Research

## Abstract

Transformer language models have demonstrated impressive generalization capabilities in natural language domains, yet we lack a fine-grained understanding of how such generalization arises. In this paper, we investigate length generalization—the ability to extrapolate from shorter to longer inputs—through the lens of *task association*. We find that length generalization can be *transferred* across related tasks. That is, training a model with a longer and related auxiliary task can lead it to generalize to unseen and longer inputs from some other target task. We demonstrate this length generalization transfer across diverse algorithmic tasks, including arithmetic operations, string transformations, and maze navigation. Our results show that transformer models can inherit generalization capabilities from similar tasks when trained jointly. Moreover, we observe similar transfer effects in pretrained language models, suggesting that pretraining equips models with reusable computational scaffolding that facilitates extrapolation in downstream settings. Finally, we provide initial mechanistic evidence that length generalization transfer correlates with the re-use of the same attention heads between the tasks. Together, our findings deepen our understanding of how transformers generalize to out-of-distribution inputs and highlight the compositional reuse of inductive structure across tasks.

## 1  Introduction

A central theme of transformer language models is their ability to generalize. By scaling up data and model size, large language models develop emergent abilities that exceed expectations [Wei et al., 2022]. They can also transfer knowledge across domains and tasks [OpenAI, 2024, Brown et al., 2020, Sanh et al., 2022]. While it is widely believed that language models are not simply parroting or memorizing their training data, we still lack a fine-grained understanding of how language models apply skills learned during training to potentially unseen problems.

The out-of-distribution (OOD) generalization capabilities of language models have garnered much attention in the literature [Anil et al., 2022, Zhang et al., 2024, Yang et al., 2024]. In this work, we study a canonical example of OOD generalization, *length generalization*, which is the ability to generalize from shorter to longer inputs [Zhou et al., 2023]. There is a long line of work focusing on improving length generalization of arithmetic tasks in transformers, which has spurred innovations in positional encoding schemes and transformer architecture [Cho et al., 2024, McLeish et al., 2024]. Closely related is the concept of compositional generalization, where the model combines previously learned skills to solve new problems [Yang et al., 2024, Xu et al., 2024].

---

[*]Corresponding author. zcai75@wisc.edu

39th Conference on Neural Information Processing Systems (NeurIPS 2025).

In this work, we study a new mechanism underlying length generalization: *extrapolation by association*. We hypothesize that, when faced with a problem outside its training distribution, language models can use related skills to solve it. Specifically, we ask: Can generalization to longer inputs in one task *transfer* to another task that is only trained on short examples?

To showcase the length generalization transfer capabilities in transformers, we choose three distinct groups of synthetic tasks. The tasks in each group are related such that they represent similar algorithmic procedures. Within each group, we train multiple tasks together, and crucially, we train an "auxiliary task" at a longer length and a "main task" at a shorter length. Using this setup, we observe that the shorter main task generalizes to the length of the longer auxiliary task when trained together. See Figure 2 for the tasks and respective lengths used in each experiment.

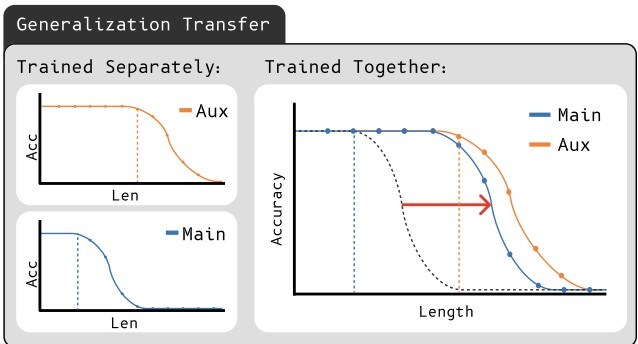

Figure 1: Trained separately, each task fails to generalize to longer inputs. When trained jointly, the main task inherits the generalization range of the auxiliary task.

**Contributions**

1. We present the phenomenon of **length generalization transfer**, in which transformer models trained on related tasks exhibit extrapolation behavior not present when trained on the target task alone, providing new insights on the effect of multitask training on length generalization.

2. We show that the same phenomenon replicates in pretrained language models, and that natural language pretraining transfers length generalization capabilities to synthetic downstream tasks.

3. We provide mechanistic evidence that transfer correlates with shared internal computation—specifically, the reuse of attention heads across tasks.

## 2 Related Works

**Length Generalization.** Length generalization concerns extrapolating to longer sequence lengths than those seen during training [Dubois et al., 2019, Hupkes et al., 2020, Newman et al., 2020, Anil et al., 2022]. Previous approaches include architectural modifications such as specialized positional embeddings [Press et al., 2021, Li et al., 2023, Ruoss et al., 2023, Kazemnejad et al., 2024, Sabbaghi et al., 2024, Cho et al., 2024, Zhou et al., 2024, McLeish et al., 2024], looping [Fan et al., 2024], novel attention mechanisms [Duan et al., 2023, Li et al., 2025], and input format augmentation [Zhou et al., 2023, 2024]. Beyond arithmetic, Yehudai et al. [2021] studies length generalization in graph tasks. In contrast, our work examines a novel mechanism from which length generalization emerges: transfer from related tasks. Finally, closely related to our work, "task hinting" [Awasthi and Gupta, 2023] trains sorting and increment-by-one tasks with simpler auxiliary tasks, showing improvements in length generalization performance.

**Compositional Capabilities.** To explain emergent capabilities in language models, many works study compositional generalization to understand whether transformers can gain abilities beyond those in the training set. Yu et al. [2023], Zhao et al. [2025] and Hosseini et al. [2024] design benchmarks testing the ability to combine learned skills to solve compositional math problems. Ahuja and Mansouri [2024] derive provable guarantees for length and compositional generalization conditioned on training set diversity. Some works use synthetic tasks to probe compositional generalization. Ramesh et al. show transformers achieve compositional generalization on unseen combinations using a series of bijections and permutations applied to strings, while Abedsoltan et al. [2025] show similar results on families of parity functions.

For the specific task of reverse addition, works like Quirke and Barez [2023] and Quirke et al. [2025] identify computational circuits responsible for compositional subtasks and show transferability of such circuits to the related task of subtraction.

# 3 Experimental Settings

**Models.** For from-scratch experiments, we use transformer models with 6 heads and 6 layers, following the Llama architecture [AI@Meta, 2024], which uses Rotary Positional Embeddings (RoPE) [Su et al., 2023] for position encoding. For experiments with pretrained models, we use SmolLM [Allal et al., 2024], which provides access to intermediate checkpoints during pretraining, allowing us to investigate how length generalization transfer evolves over time.

**Tasks.** We evaluate length generalization transfer across three categories of algorithmic problems: arithmetic, string manipulation, and maze solving. Our tasks include:

- **Arithmetic Tasks**
    - `reverse add` – Compute the sum of two integers, presented in reversed order.
    - `no carry` – Compute digit-wise sums mod 10, without carry propagation.
    - `carry only` – Output a binary mask indicating carry positions during addition.
    - `reverse subtract` – Compute the reversed digit-wise difference between two numbers.
    - $n \times 3$ `CoT multiply` – Multiply an $n$-digit number by 3, with chain-of-thought steps.
- **String Manipulation Tasks**
    - `string copy` – Return the input string unchanged.
    - `MQAR` (Multi-Query Associative Recall) [Arora et al., 2023] – Given a repeated query substring, retrieve the next character following each occurrence.
    - `capitalize` – Flip the case of all alphabetic characters (lower $\leftrightarrow$ upper).
    - `reverse` – Reverse the character order of the input string.
    - `capitalize-reverse` – Apply both reversal and case-flipping to the input string.
- **Maze Tasks**
    - `DFS trace` – Simulate a depth-first search from a start node to a goal node in a maze.
    - `shortest path` – Return the optimal (shortest) path between a start and goal node.

**Task Groups.** We construct *task groups* by pairing a main task, trained on short sequence lengths, with one or more auxiliary tasks, trained on longer sequences. The main goal is to evaluate whether training on a related auxiliary task improves the main task's ability to generalize to longer inputs, despite never seeing such lengths during training. The list of task groups are:

| Main Task (Train Length) | Auxiliary Task(s) (Train Length) |
|---|---|
| `reverse add` (16) | `no carry` & `carry only` (32) |
| `reverse add` (16) | `reverse subtract` (32) |
| `reverse add` (8) | $n \times 3$ `CoT multiply` (16) |
| `string copy` (16) | `MQAR` (32) |
| `capitalize-reverse` (16) | `capitalize` (32), `reverse` (32) |
| `DFS trace` (32) | `shortest path` (64) |

**Data sampling and Task Length.** Since we train under a multi-task setting, at each iteration, a task is sampled uniformly at random from a predefined task group. For the selected task, an individual training example is constructed based on a single governing parameter: *length*, which determines the size or complexity of the problem instance. The length of each example is sampled uniformly from a specified range for that task. All training data is generated on-the-fly during training.

Since the notion of *length* varies across task types, we define length for each task as:

- **Addition Tasks**: the maximum number of digits in both operands.
- **String Tasks**: the number of characters in the input string.
- **Maze Tasks**: the number of nodes in the input maze graph. See Section 4.3 for further details.

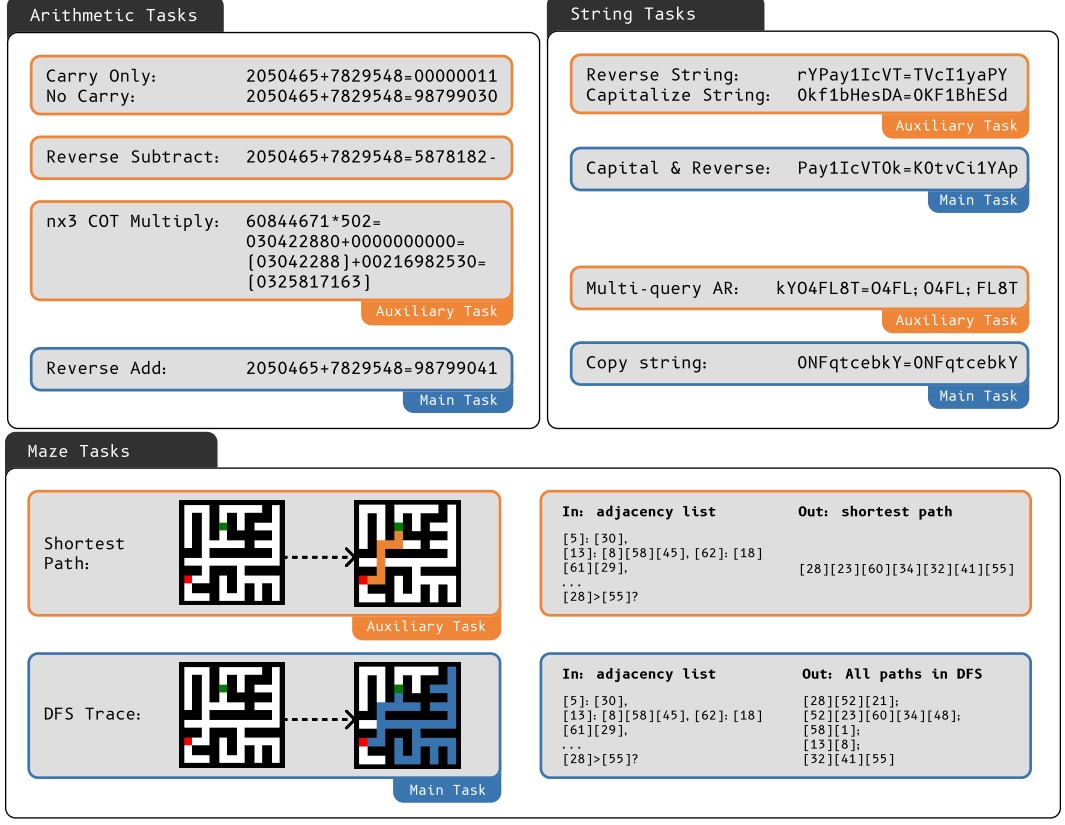

Figure 2: Overview of the tasks used in our length generalization transfer experiments, spanning three domains: arithmetic, string manipulation, and maze solving. Each group consists of a main task trained on shorter sequences and one or more auxiliary tasks trained on longer ones. We study whether generalization to longer inputs can be transferred from the auxiliary to the main task.

**Training and Evaluation.** Each example consists of an input-output pair. We use a loss mask to train only on output tokens (and for MQAR, only on answer characters). At test time, we evaluate using exact match accuracy on a fixed test set of 1024 examples. For each configuration, we report results across 5 random initialization seeds but the dataset is kept the same. Full experimental configurations and hyperparameter details are provided in Appendix C.

## 4   Length Generalization Transfer in Algorithmic Tasks

In this section, we demonstrate that while length generalization is often difficult for algorithmic tasks, it can emerge through transfer when the model is co-trained on longer auxiliary tasks. Figure 2 illustrates the three categories of tasks we study—arithmetic operations, string transformations, and maze navigation.

### 4.1   Arithmetic Tasks

Reverse addition has become a popular synthetic task for studying length generalization [Lee et al., 2023, Shen et al., 2023, Zhou et al., 2023, 2024, Cho et al., 2024, McLeish et al., 2024, Lee et al., 2025] in Transformers. The task involves calculating the sum of two randomly sampled integers, and length generalization in this task involves training on examples up to some fixed length, and generalizing on test data beyond the training lengths. Here, we adopt the `reverse add` format proposed by Lee et al. [2023], where the operands and the sum are reversed for faster learning. For the auxiliary tasks, we consider (1) `reverse subtract`, which computes the difference between

two operands, (2) `no carry`, which computes the digit-wise sum mod 10, ignoring the carries, and (3) `carry only`, which computes the locations where a carry happens in the addition.

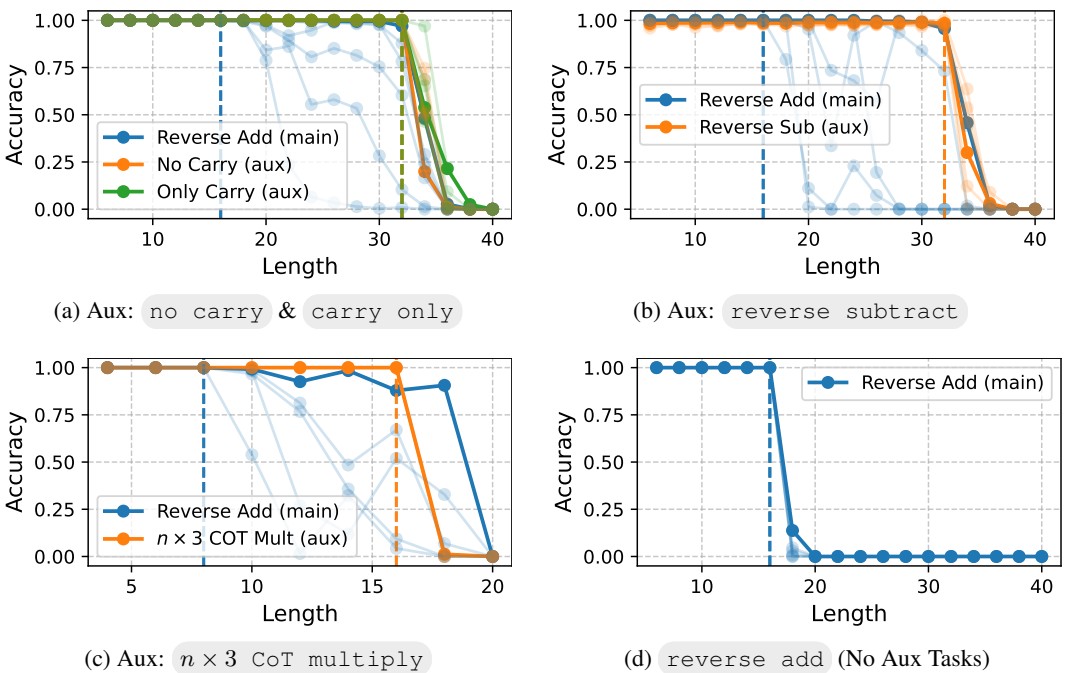

(a) Aux: `no carry` & `carry only`

(b) Aux: `reverse subtract`

(c) Aux: $n \times 3$ `CoT multiply`

(d) `reverse add` (No Aux Tasks)

Figure 3: Length generalization results for addition-related task groups. The main task is `reverse add`, with performance shown when trained with different auxiliary tasks. Each model is trained with 5 random seeds; best-performing runs are shown in bold. The dashed vertical line indicates the maximum training length for each task. When trained alone (d), the model fails to generalize beyond training length. Co-training with related auxiliary tasks (a-c) enables extrapolation to longer inputs.

As shown in Figure 3, models trained only on `reverse add` (Figure 3d) struggle to generalize beyond the training length. However, when co-trained with longer auxiliary tasks (Figures 3a, 3b, 3c), the model successfully extrapolates, often matching the auxiliary task's generalization range. This provides empirical evidence that length generalization can transfer across tasks.

It is worth noting that the generalization behavior is not entirely robust: different random seeds yield noticeably different outcomes, suggesting unstable training dynamics. We discuss this instability further in Section 6.2.

## 4.2 String Tasks

We now turn to string operations, where we observe similar transfer effects on two task groups. The tasks include: `string copy`, which returns the input unchanged; `MQAR` (Multi-Query Associative Recall) [Arora et al., 2023], where the model retrieves the next character given a random substring; `reverse`, which reverses character order; `capitalize`, which inverts letter case; and `capitalize-reverse`, combining case inversion and reversal.

Figure 4 shows that when trained on main tasks alone (Figures 4b, 4d), the model does not generalize beyond the training range. On the other hand, Adding training with auxiliary tasks enables substantial extrapolation (as shown in Figures 4a and 4c).

## 4.3 Maze Tasks

Lastly, we examine maze-solving tasks as a testbed for length generalization transfer. We define a maze as a spanning tree over a square grid, generated using Wilson's algorithm [Wilson, 1996], which

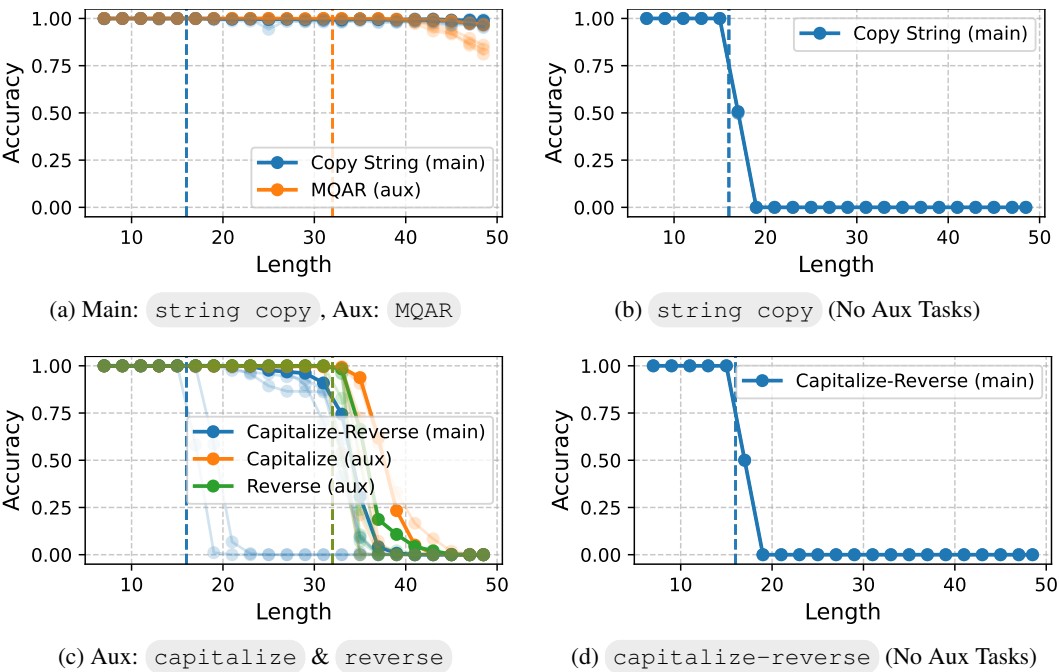

(a) Main: `string copy` , Aux: `MQAR`

(b) `string copy` (No Aux Tasks)

(c) Aux: `capitalize` & `reverse`

(d) `capitalize-reverse` (No Aux Tasks)

Figure 4: Performance plots for string tasks. When trained alone (b, d), models fail to generalize beyond their training range. Co-training with auxiliary tasks (a, c) enables substantial length extrapolation.

ensures uniform sampling via loop-erased random walks. For each problem instance, we randomly sample a start and end node, and the model is tasked with producing a path from start to end. Mazes are represented as adjacency lists, with each node and its neighbors encoded as individual tokens (e.g., [1], [2], ..., [64]). Input/output formatting examples are shown in Figure 2 and Section C.2.

A challenge in defining length generalization for mazes is that increasing grid size introduces unseen node tokens at test time. To avoid this, we fix the grid size and instead vary the number of nodes included in the spanning tree. Specifically, we define the input length as the total number of nodes in the maze graph and generate partial mazes by stopping Wilson's algorithm early. For example, to construct a 32-node maze on an $8 \times 8$ grid, we run the algorithm until 32 nodes are added. The resulting maze may not span the full grid but remains a valid traversal problem. Figure 5 illustrates such partial mazes with 16, 32, and 64 nodes.

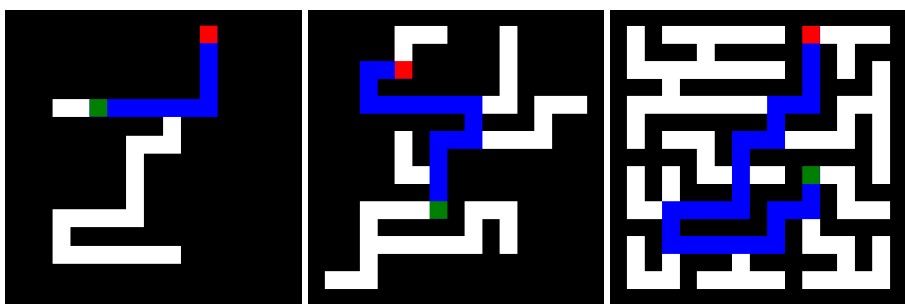

Figure 5: $8 \times 8$ mazes with number of nodes equal to 16, 32, and 64. We define length generalization as the ability to generalize to mazes with a higher number of nodes.

We consider two maze tasks: (1) `shortest path` , where the model outputs the shortest path from start to end node, and (2) `DFS trace` , where the model simulates a depth-first search traversal (including backtracking). Shortest path is harder to learn perfectly, as it requires "lookahead" at branch points, while DFS trace allows exploration and backtracking. Figure 6 shows that in the

multi-task setting, the addition of `shortest path` helps `DFS trace` generalize to higher lengths. The opposite is true as well: `DFS trace` helps `shortest path` generalize to higher lengths, which is shown in Figure 7.

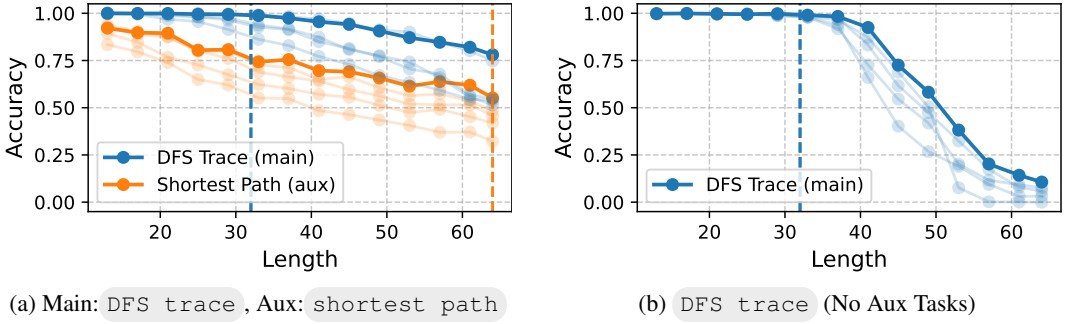

(a) Main: `DFS trace`, Aux: `shortest path`        (b) `DFS trace` (No Aux Tasks)

Figure 6: Performance plots for maze tasks. Co-training `DFS trace` with `shortest path` (a) enables generalization to longer lengths compared to training on `DFS trace` alone (b).

### 4.3.1 Transfer with Swapped Main and Auxiliary Tasks

We consider another maze task group where we the main and auxiliary tasks are reversed relative to Section 4.3. In this case, the main task is `shortest path`, and the auxiliary task is `DFS trace`. As shown in Figure 7, co-training with the auxiliary task again improves length generalization performance. While `shortest path` is more difficult than `DFS trace`, the model benefits from learning a related traversal strategy.

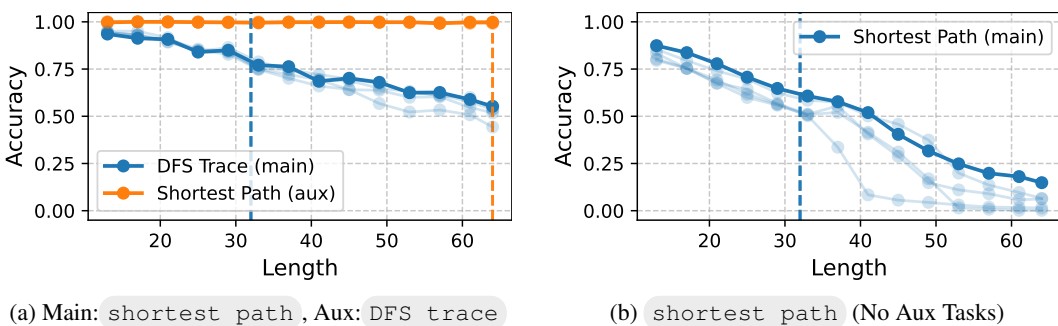

(a) Main: `shortest path`, Aux: `DFS trace`        (b) `shortest path` (No Aux Tasks)

Figure 7: Length generalization results for maze task group with reversed task roles. Co-training `shortest path` with `DFS trace` (a) leads to improved generalization over training on `shortest path` alone (b).

### 4.4 Control Tasks

To verify that length generalization transfer does not arise from merely seeing longer inputs, we further test arithmetic tasks and string operations with control auxiliary tasks. or arithmetic, we use `copy-first-op`, which follows the addition format but simply copies the first operand. For string operations, we pair `string copy` with `reverse`. As expected, length generalization transfer is not observed with unrelated task (Figure 8).

## 5 Length Generalization Transfer from Pretraining

Remarkably, we find that natural language pretraining can serve as an effective form of *implicit auxiliary task* that enhances length generalization in synthetic tasks. To explore this, we finetune various checkpoints of SmolLM-360M [Allal et al., 2024] on `reverse add` and `shortest path`

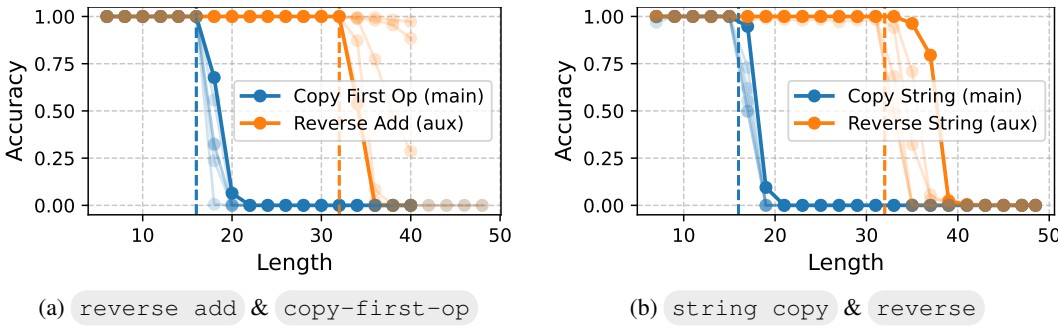

(a) `reverse add` & `copy-first-op`    (b) `string copy` & `reverse`

Figure 8: Control tasks for (a) addition and (b) string operations. These unrelated task pairs fail to produce length generalization transfer, confirming that task relatedness is crucial.

tasks. SmolLM is released by Huggingface and pretrained on a diverse corpus containing natural language and programming data, which includes long-range structures and dependencies.

Before finetuning, we verify that the model does not already solve these tasks. For `reverse add`, a zero-shot evaluation using prompt-based input results in near-zero accuracy, confirming that the model has not learned this task during pretraining. For the maze task, all node tokens are newly introduced during finetuning, meaning the entire input format is unseen by the pretrained model.

We then finetune models from multiple publicly available checkpoints, taken throughout the pretraining process (from step 160K to 2.56M), and evaluate their length generalization performance on out-of-distribution inputs. As shown in Figure 9, we observe a clear trend: generalization to longer inputs improves steadily with pretraining progress, for both arithmetic and maze-solving tasks. This suggests that natural language pretraining instills reusable inductive biases that transfer to novel tasks—even when those tasks have little structural resemblance to natural language. We speculate the extent of generalization transfer from pretrained models may not be limited to length generalization, but could extend to other forms of out-of-distribution generalization such as compositional reasoning, distributional shifts, and task complexity. Future work could explore whether similar transfer effects exist for other generalization challenges.

Additionally, we confirm that length generalization transfer is not limited to small models trained from scratch, but also emerges in finetuned pretrained models. Additional results across other task groups are provided in Appendix A.3.

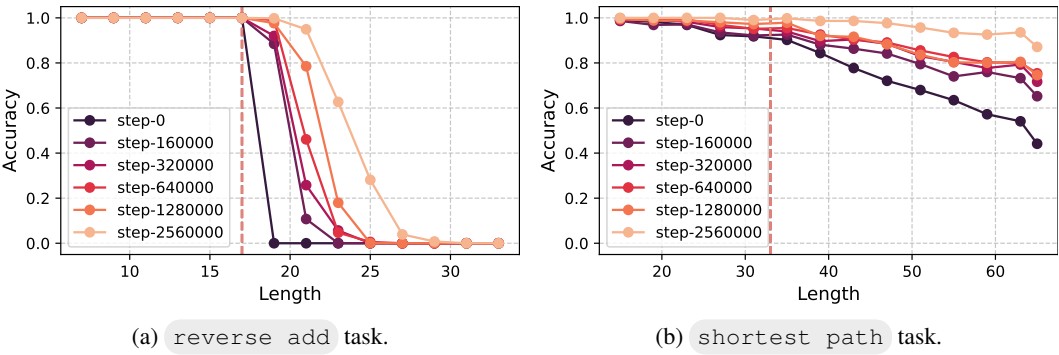

(a) `reverse add` task.    (b) `shortest path` task.

Figure 9: Finetuning at different SmolLM-360M checkpoints reveals that length generalization transfer improves with more natural language pretraining.

## 6 Ablations

In this section, we present several complementary analyses to better understand the conditions under which transfer occurs. We examine the effect of varying the length configurations of the main and auxiliary tasks and also provide an initial mechanistic explanation of the transfer phenomenon based

on circuit sharing between tasks. Additional analyses, including the instability of training dynamics (Section 6.2) and the effect of positional encodings (Section 6.3) are included in the Appendix.

## 6.1 Varying Main and Auxiliary Task Lengths

In our previous experiments, we fixed the main task length to 16 and the auxiliary task length to 32. A natural question is: does length generalization transfer persist across other main–auxiliary length configurations? To investigate this, we define the *generalization gap* (Figure 10), a scalar between 0 and 1 that quantifies the discrepancy in performance between the main and auxiliary tasks across a range of evaluation lengths. A smaller generalization gap indicates stronger transfer, with a value of 0 implying perfect alignment between the main and auxiliary generalization curves.

First we fix the task group `reverse add`, `no carry` and `carry only`. Then, we systematically vary the training lengths of both main and auxiliary tasks across the range $\{4, 8, 16, \ldots, 256\}$ and compute the average generalization gap over three random seeds. As shown in Figure 10, we find that the transfer effect is most effective when the ratio between the auxiliary and main lengths is between 0.5 and 2. The intuitive explanation is that, when the difference between task length is too high, the model will overfit to the task length difference and therefore do not exhibit length transfer.

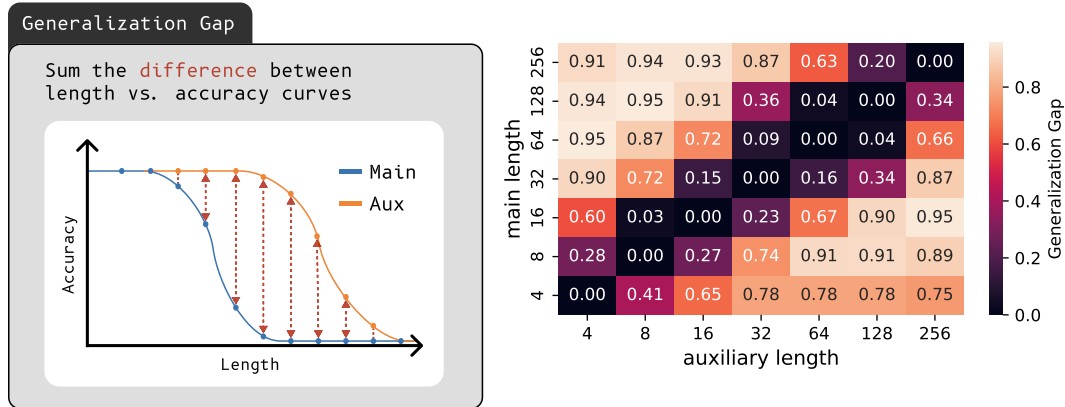

Figure 10: (a) The *generalization gap* is defined as the average difference in accuracy between the main and auxiliary tasks across evaluation lengths, normalized to the range [0, 1]. A lower value indicates better transfer. (b) Generalization gap across different combinations of main (`reverse add`) and auxiliary (`no carry` & `carry only`) training lengths. The transfer effect is strongest when the ratio between auxiliary and main lengths is between 0.5 and 2, as shown by the dark diagonal band.

## 6.2 Unstable Training Dynamics in Length Generalization Transfer

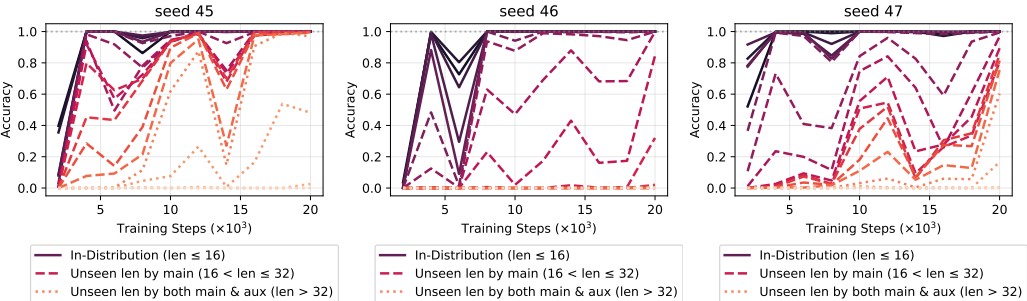

Figure 11: Training curves for the `reverse add` when co-trained with `no carry` and `carry only`. Accuracy in the transfer region (length 17–32) fluctuates significantly, illustrating unstable training dynamics in length generalization transfer.

As shown in Figures 3, 4, and 6, not all random seeds exhibit successful length generalization transfer. In our experiments with 5 different seeds per task group, we observe considerable variability in length generalization transfer performance. The variability is entirely due to different model initializations, since we keep the dataset the same between runs. To better illustrate this instability, we visualize training dynamics in Figure 11.

The plots show training curves for the `reverse add` main task when co-trained with `no carry` and `carry only` auxiliary tasks. During evaluation, we sweep over input lengths from 1 to 36, which is classified into three regimes:

- In-distribution (length 1–16): These inputs fall within the training range for the main task. Accuracy in this regime improves quickly and remains stable.
- Expected transfer range (length 17–32): These inputs are unseen by the main task but seen by the auxiliary tasks. Performance in this range is highly variable and sensitive to training dynamics.
- Fully OOD (length >32): These inputs are unseen by both the main and auxiliary tasks. As expected, accuracy in this regime remains low.

### 6.3 Rotary Position Encoding Encourages Length Generalization Transfer

In length generalization literature, *NoPE* (no positional encoding) is often favored for its strong extrapolation on individual tasks. However, many modern transformer models use *Rotary Positional Encoding* (RoPE) due to its empirical robustness in long-context and real-world settings [Peng et al., 2023, Ding et al., 2024, Barbero et al., 2024].

We re-evaluate our multitask transfer setup under both encoding schemes. Across task families, RoPE consistently yields stronger length-generalization *transfer* from auxiliary to main tasks. Detailed per-task curves are provided in Appendix A.1. Figure 12 summarizes the overall trend. This finding is orthogonal to the previous understanding that NoPE is better suited for length generalization and potentially explains the superior performance of RoPE in real-world models and tasks.

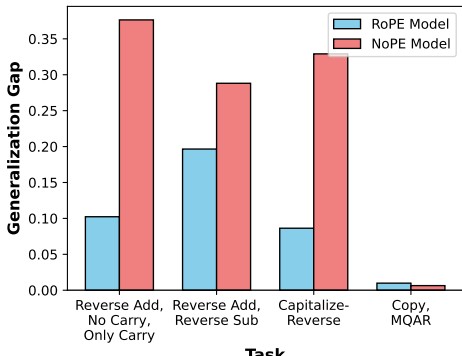

Figure 12: Comparison of generalization gap across task groups. Smaller gap means stronger transfer. RoPE consistently outperforms NoPE, indicating that rotary embeddings better support cross-task extrapolation.

## 7 Limitations

While our work demonstrates length generalization transfer across a range of synthetic tasks, several important limitations remain. First, our study does not provide a formal theoretical framework for understanding when and why transfer occurs. Without a principled understanding of the underlying mechanisms, predicting or optimizing transfer remains challenging. Second, our experiments are limited to relatively simple algorithmic domains with well-defined length parameters and deterministic solution paths. While this setup allows for controlled comparisons, it is unclear whether similar transfer effects would hold in settings that involve hierarchical reasoning, abstract problem-solving, or tasks requiring integration of multiple skills simultaneously. Addressing these limitations is a promising direction for future work and could further illuminate the generalization capabilities of transformer models in more realistic settings.

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

# A Additional Results

## A.1 Detailed Plots for Rotary vs. NoPE Models

To complement Section 6.3, we show full length-generalization curves for the *No Positional Encoding* (NoPE) and *Rotary Positional Encoding* (RoPE) variants under the same task settings. Each subplot reports exact-match accuracy versus input length. The dashed vertical line indicates the maximum training length for each task. In all domains, RoPE models exhibit smoother extrapolation and better alignment between main and auxiliary tasks.

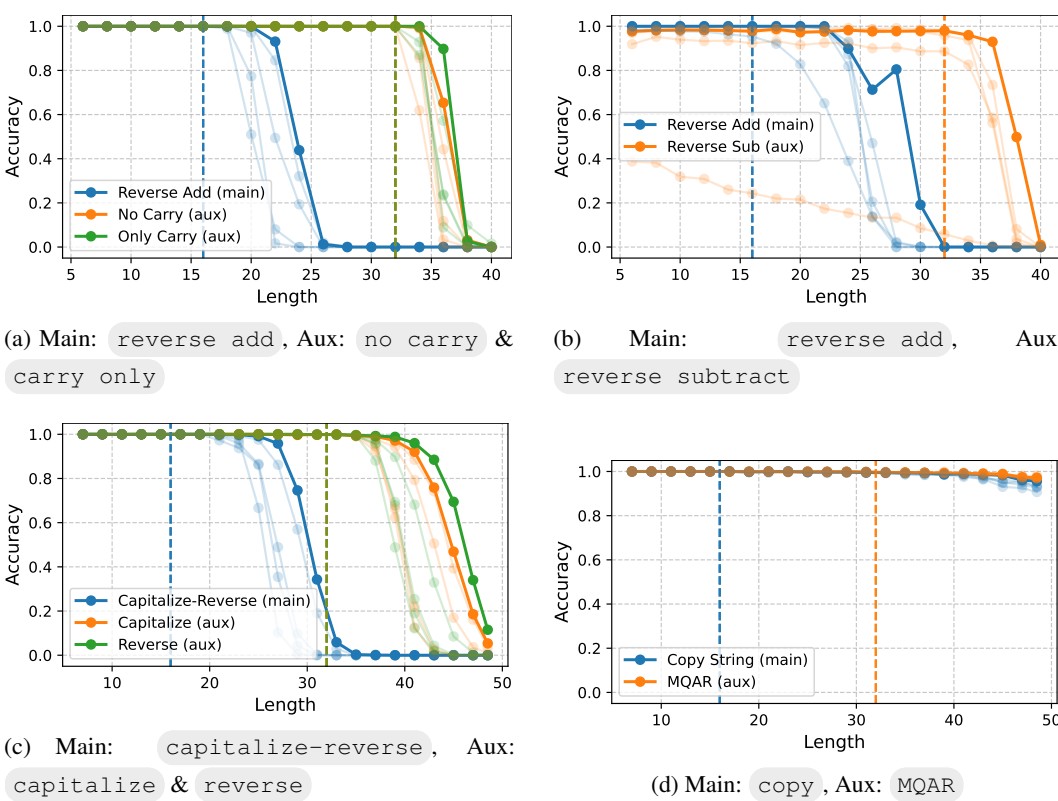

(a) Main: `reverse add`, Aux: `no carry` & `carry only`

(b) Main: `reverse add`, Aux: `reverse subtract`

(c) Main: `capitalize-reverse`, Aux: `capitalize` & `reverse`

(d) Main: `copy`, Aux: `MQAR`

Figure 13: Detailed performance curves comparing RoPE and NoPE variants across arithmetic and string tasks. RoPE models maintain strong transfer to longer lengths, while NoPE variants degrade rapidly beyond the training range.

## A.2 Additional Results on Arithmetic and String Tasks

For task groups with two auxiliary tasks– `reverse add` with `no carry` and `carry only`, and `capitalize-reverse` with `capitalize` and `reverse` –we additionally evaluate the effect of training with only one of the auxiliary tasks. As shown in Figure 14, length generalization transfer performance consistently declines when only a single auxiliary task is used, compared to co-training with both. Notably, the choice of auxiliary task matters: models trained with the more relevant auxiliary (`no carry` or `reverse`) exhibit stronger generalization than those trained with less relevant ones (`carry only` or `capitalize`). These results reinforce the importance of task alignment for successful transfer. As shown in Figure 14, length generalization transfer performance consistently declines when only a single auxiliary task is used, compared to co-training with both. Notably, the choice of auxiliary task matters: models trained with the more relevant auxiliary (`no carry` or `reverse`) exhibit stronger generalization than those trained with less relevant ones (`carry only` or `capitalize`). These results reinforce the importance of task alignment for successful transfer.

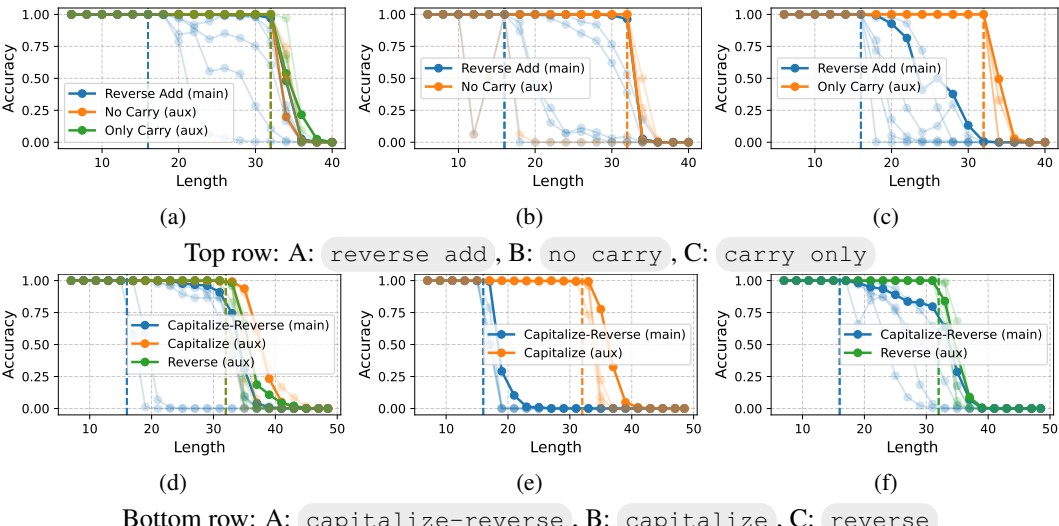

Top row: A: `reverse add`, B: `no carry`, C: `carry only`

Bottom row: A: `capitalize-reverse`, B: `capitalize`, C: `reverse`

Figure 14: Additional results for arithmetic and string (copy) task groups. Each row shows performance on the main task (A) when co-trained with: both auxiliary tasks (left), only one of the auxiliary task (middle & right). Performance degrades when training with only one auxiliary task, especially when the auxiliary is less structurally aligned with the main task.

## A.3 Finetuning from Pretrained Models

We replicate our length generalization transfer experiments using a pretrained language model, SmolLM-360M, where we observe similar patterns of length generalization transfer as in the from-scratch setting. Figure 15 presents results across three arithmetic task groups and one string manipulation group. As with our earlier experiments, co-training with structurally related auxiliary tasks facilitates generalization beyond the training length. Notably, we also confirm that control task pairs–such as `reverse add` with `copy-first-op` –do not lead to successful transfer. Orthogonal to the length generalization transfer, results show that SmolLM-360M exhibits strong inherent generalization in copying tasks (15c, 15d).

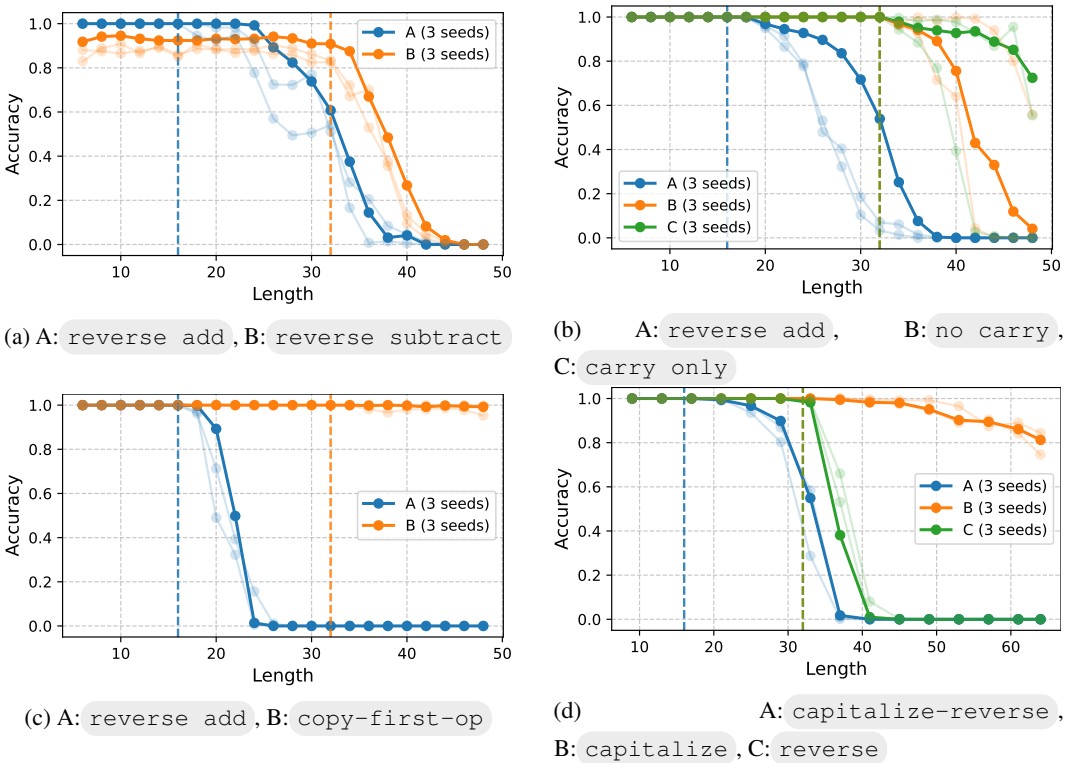

(a) A: `reverse add`, B: `reverse subtract`

(b) A: `reverse add`, B: `no carry`, C: `carry only`

(c) A: `reverse add`, B: `copy-first-op`

(d)     A: `capitalize-reverse`, B: `capitalize`, C: `reverse`

Figure 15: Length generalization transfer with the pretrained model SmolLM-360M. (a–c): Arithmetic task groups. In (a) and (b), we observe successful transfer from auxiliary to main tasks, mirroring results from from-scratch training. In (c), no transfer occurs when using the control task `copy-first-op`, confirming the importance of task relevance. (d): String manipulation task, showing transfer from `capitalize` and `reverse` to `capitalize-reverse`. Overall, the transfer effect persists in the pretrained model.

# B  Mechanistic Evidence of Circuit Sharing

In this section we consolidate the mechanistic analysis of *length generalization transfer*, showing that successful transfer coincides with reuse of internal attention circuits across related tasks.

**Metrics and protocol.**    We study whether transformer models reuse similar attention mechanisms across tasks when length generalization transfer occurs. We use two complementary metrics:

- **Attention matrix difference:** sum of entry-wise absolute differences between attention matrices (per head) for two tasks. Lower values indicate more similar attention patterns.

- **Attention-head mean-ablation map difference:** for each head (6 layers $\times$ 6 heads), we replace its output with the batch mean and measure the accuracy drop (activation patching). This produces a head-importance map per task; we then take the average absolute difference between the two maps. Lower values indicate more similar head usage.

This follows standard activation-patching methodology used in mechanistic-interpretability studies [Wang et al., 2022, Cammarata et al., 2021, Olsson et al., 2022]. Across checkpoints, reductions in the *generalization gap* (defined in Fig. 10) generally coincide with smaller differences in both metrics—i.e., tighter alignment of attention mechanisms across tasks when transfer strengthens.

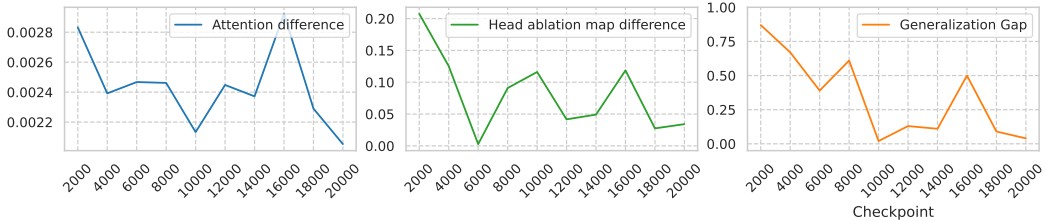

Figure 16:  Arithmetic task group ( `reverse add` with auxiliaries `reverse subtract` / `no carry` ). Evolution of generalization gap, attention-matrix difference, and head mean-ablation map difference across checkpoints. When transfer improves (smaller gap), attention mechanisms align (smaller differences).

### B.0.1   Example Attention-Head Ablation Maps

We visualize the *attention-head mean-ablation maps* for a pair of related tasks— `reverse add` and `reverse subtract` —across four training checkpoints (Figure 17). Each $6 \times 6$ matrix represents the importance of each attention head: the value at position $(i, j)$ indicates the drop in accuracy when head $i$ in layer $j$ is replaced with the mean activation across the batch. These matrices reveal which heads are functionally critical for each task. If two tasks reuse the same circuitry, their ablation maps will appear similar; our scalar similarity metric is the average absolute difference between the two matrices.

### B.0.2   Extended Results Across Tasks

We next compare how the two circuit-similarity metrics track with the generalization gap across training checkpoints for string, arithmetic, and control tasks.

**String tasks.**    Figure 18 shows that the raw attention-matrix difference does not correlate well with generalization for string tasks, whereas the ablation-map difference does. Shared head usage, rather than raw attention weights, better captures functional similarity.

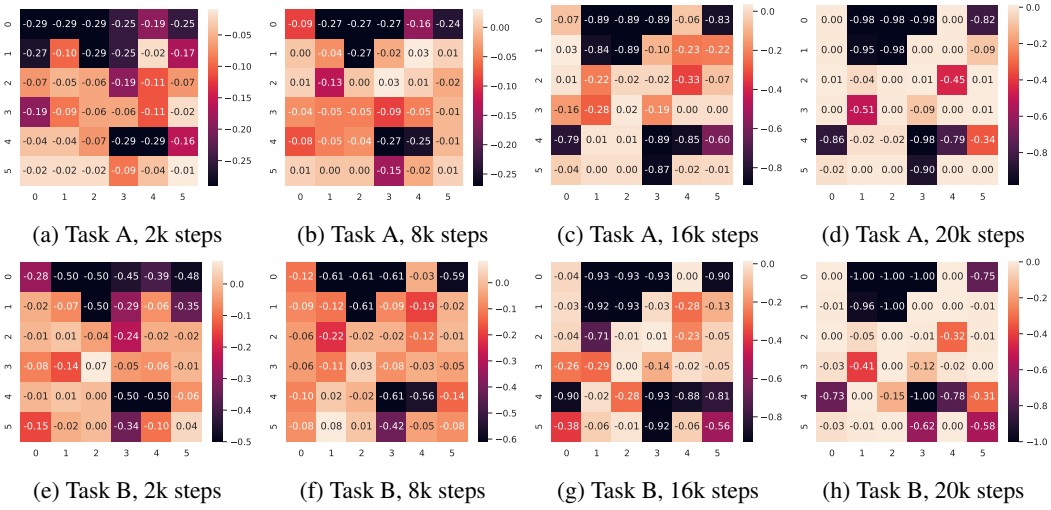

Figure 17: Mean-ablation maps for `reverse add` and `reverse subtract` across checkpoints. Each $(i, j)$ entry shows the accuracy drop after mean-ablating head $i$ in layer $j$. Similar maps indicate overlapping computational circuits.

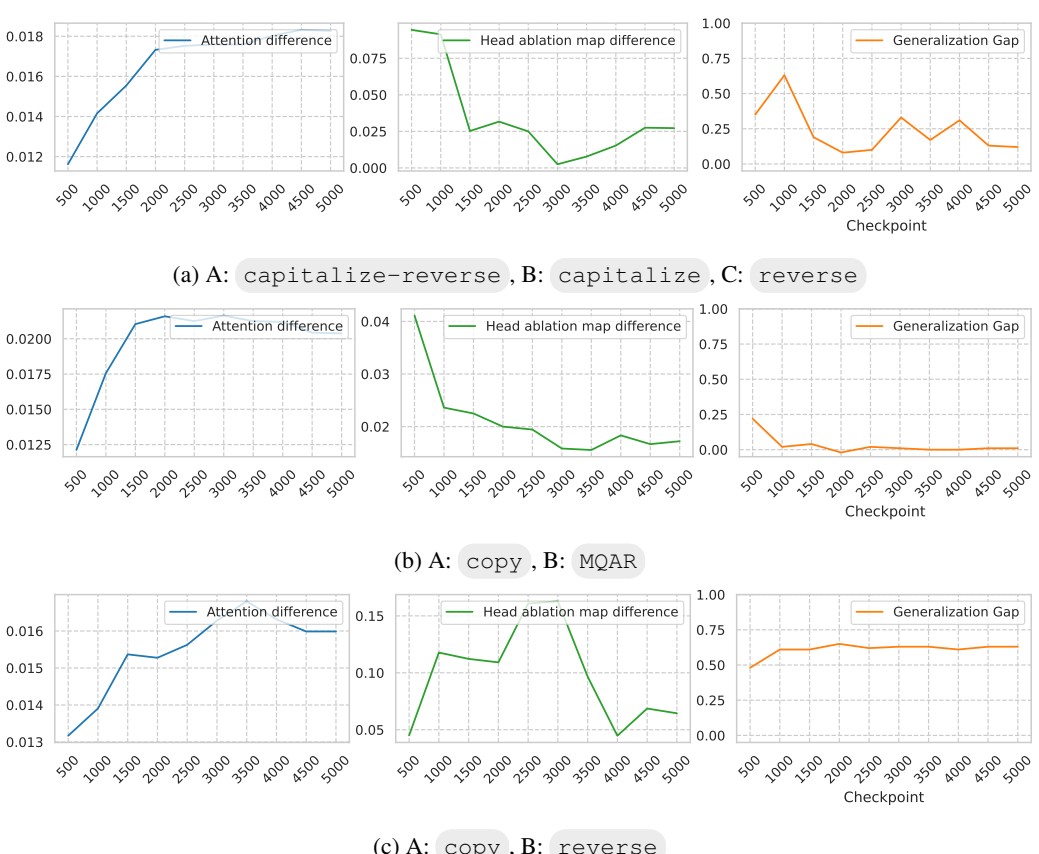

Figure 18: Circuit-sharing results for string task pairs. The attention-matrix difference shows weak correlation with generalization, whereas the head-ablation map difference tracks it closely, highlighting shared head usage.

**Arithmetic tasks.** For arithmetic task pairs (Figure 19), both metrics strongly correlate with the generalization gap, suggesting that these tasks share not only head usage but also detailed attention-pattern structure.

**Control tasks.** For unrelated task pairs such as `reverse add` with `copy-first-op`, neither metric correlates with performance, confirming that the observed correlations are not incidental.

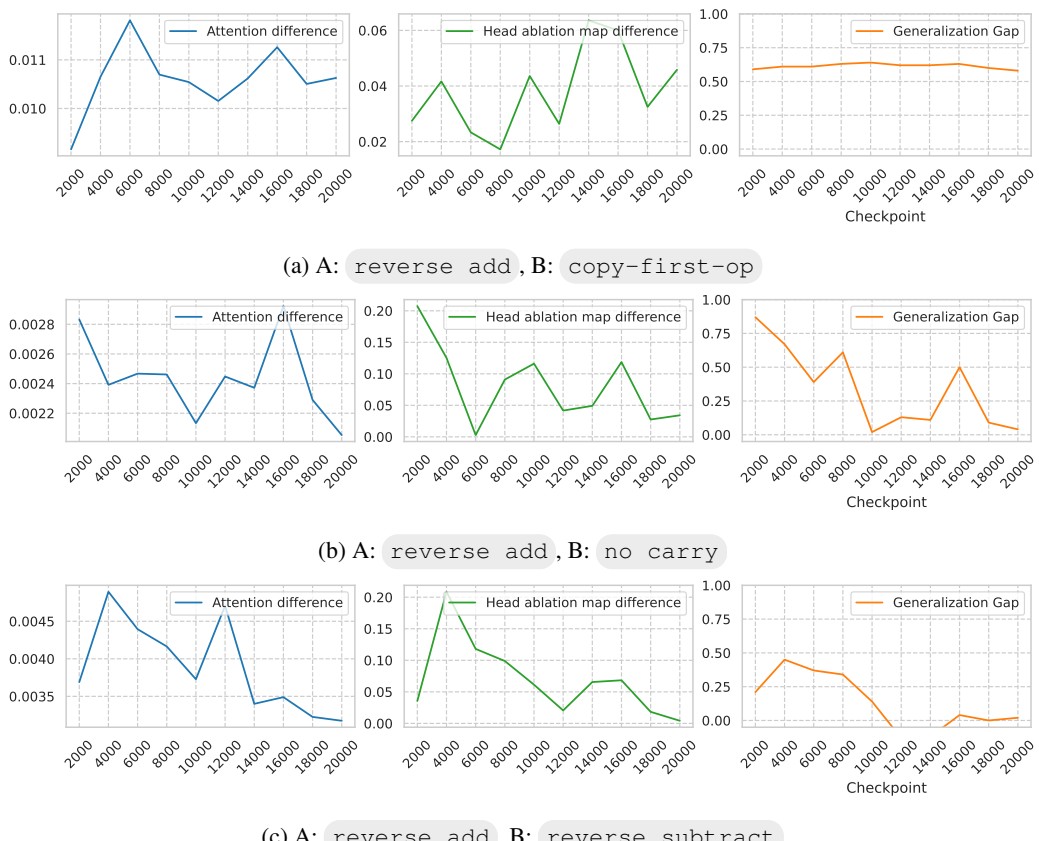

(a) A: `reverse add`, B: `copy-first-op`

(b) A: `reverse add`, B: `no carry`

(c) A: `reverse add`, B: `reverse subtract`

Figure 19: Circuit-sharing results for arithmetic tasks. Both attention-matrix and head-ablation map differences correlate with generalization gap in related task pairs (b,c) but not in the unrelated control pair (a).

**Takeaway.** Across arithmetic, string, and maze domains (not shown), strong length-generalization transfer coincides with shared attention-head usage between tasks. This supports the hypothesis that transformers reuse a compositional "scaffold" of attention circuits when transferring extrapolation behavior across related problems.

# C    Experiment Details

## C.1    Model

For all experiments, we use decoder-only transformer models following the Llama architecture. Unless otherwise specified, we use Rotary Positional Embeddings (RoPE) for positional encoding; exceptions are noted in the ablation studies in Section 6.3.

For pretrained model experiments, we use SmolLM-360M [Allal et al., 2024], a compact transformer trained on natural language and code. Table 1 summarizes the model configurations used in our experiments.

Table 1: Model configurations used in our experiments.

| Model | Self-Attn Layers | Num Heads | Embedding Dim |
|---|---|---|---|
| From-Scratch | 6 | 6 | 384 |
| SmolLM | 32 | 15 | 2560 |

## C.2    Data Formats and Data Sampling

We provide examples of each task in Table 2. For all arithmetic tasks, both the inputs and outputs are written in reverse digit order. For the `n × 3 CoT multiply` task, the output includes intermediate steps where the first operand is multiplied by each digit of the second operand.

For maze-based tasks, we serialize graphs using an adjacency list format with unique node tokens, followed by a query specifying the start and end node. A detailed example is shown in Figure 20.

Table 2: Examples of algorithmic tasks used in our experiments.

| Task Name | Input | Output |
|---|---|---|
| `only carry` | 82050465+23782955= | 010010111 |
| `no carry` | 82050465+23782955= | 057323100 |
| `reverse add` | 82050465+23782955= | 067333211 |
| `reverse subtract` | 82050465+23782955= | 692674000 |
| $n \times 3$ `CoT multiply` | 60844671*502= | 030422880+0000000000= 03042288+00216982530= 0325817163 |
| `copy string` | fVOBA1fR= | fVOBA1fR |
| `Multi-Query Associative Recall` | fVOBA1fR= | fVOB;OBA1; |
| `string reverse` | fVOBA1fR= | Rf1ABOVf |
| `capitalize` | fVOBA1fR= | Fvoba1Fr |
| `capitalize-reverse` | fVOBA1fR= | rF1abovF |
| `Shortest Path` | [0]:[10], [15]:[4][5], [11]:[1][3][5], [3]:[11], [4]:[2][15], [14]:[9][5], [10]:[0][9][13], [2]:[4], [1]:[11], [7]:[5], [13]:[8][10], [5]:[11][7][14][15], [12]:[8][6], [9]:[10][14], [8]:[12][13], [6]:[12] ?[12]>[2]? | [12][8][13] [10][9][14] [5][15][4][2] |
| `DFS trace` | [0]:[10], [15]:[4][5], [11]:[1][3][5], [3]:[11], [4]:[2][15], [14]:[9][5], [10]:[0][9][13], [2]:[4], [1]:[11], [7]:[5], [13]:[8][10], [5]:[11][7][14][15], [12]:[8][6], [9]:[10][14], [8]:[12][13], [6]:[12] ?[12]>[2]? | [12][6]; [12][8][13][10][9][14][5][11][1]; [11][3]; [5][15][4][2] |

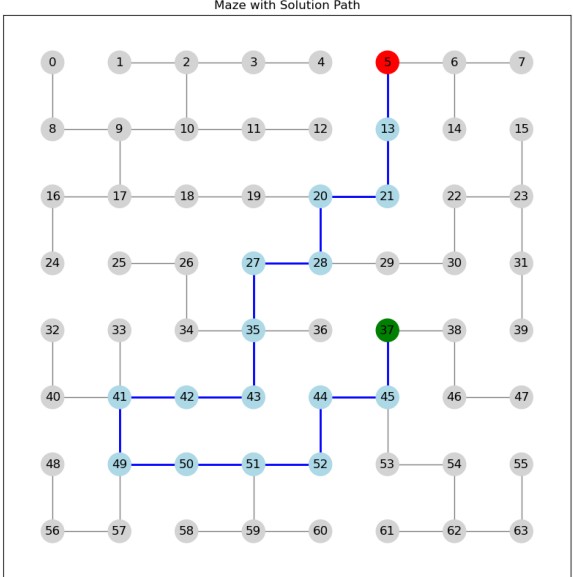

Figure 20: Detailed example of maze data format. Each node is a random number selected from $n \times n$ nodes in the grid.

## C.3 Experimental Settings

### C.3.1 Hyperparameter Configurations

Table 3 lists the hyperparameters used for training across different task domains and model types. From-scratch models are trained with a higher learning rate and larger batch sizes, while pretrained models (SmolLM-360M) use lower learning rates and shorter training schedules. All models are optimized using AdamW with a learning rate schedule that includes a warm-up phase, a constant phase, and a cosine decay phase.

Table 3: Hyperparameters for training

| Task | Batch Size | LR | Iterations | Warmup Iter | Decay Iter |
|------|-----------|-----|-----------|-------------|------------|
| Arithmetic Tasks | 1024 | 1e-3 | 20000 | 2000 | 5000 |
| String Tasks | 1024 | 1e-3 | 5000 | 500 | 1000 |
| Maze Tasks | 256 | 1e-3 | 20000 | 2000 | 5000 |
| Arithmetic Tasks (SmolLM) | 128 | 5e-5 | 2500 | 250 | 500 |
| String Tasks (SmolLM) | 128 | 5e-5 | 1000 | 100 | 500 |
| Maze Tasks (SmolLM) | 256 | 5e-5 | 2500 | 250 | 500 |

### C.3.2 Computational Resources

For all experiments in the paper, we run on a single machine with two NVIDIA GeForce RTX 3090 graphics cards. For all experiment settings, each individual training run is at most 2 hours. The total estimate of compute used, in terms of hours on the 2-GPU machine, is around 300 hours.

