# OpenReview forum: "Extrapolation by Association: Length Generalization Transfer In Transformers"
_NeurIPS.cc/2025/Conference — NeurIPS 2025 spotlight_

### Official Review · Reviewer_BHPD · 2025-07-01

**Clarity:** 3
**Significance:** 2
**Originality:** 3
**Rating:** 4
**Confidence:** 4

**Summary:**

This paper addresses the length generalization problem, a challenge facing transformers. The paper finds that length generalization can transfer between tasks: when training two tasks at different input lengths, the model may end up generalizing to the longer length on both tasks.

**Questions:**

Questions:
* see Weakness #3
* In Figure 3c, it’s presupposed that when training Reverse Add up to length 8 on its own, it won’t length-generalize (which is not shown in Figure 3d). Is this the case?
* The paper shows transfer for related tasks, but not lack of transfer in the case of unrelated tasks. Have the authors tried to pair up unrelated or less-related tasks to show that indeed “related” tasks (line 6) are needed for transfer to happen?

Minor:
* line 245: Figure 10 → Figure 9
* line 180, “or” → “For”

**Ethical Concerns:**

["NO or VERY MINOR ethics concerns only"]

**Final Justification:**

The first weakness is only partly addressed. Theoretical understanding is declared out of scope by the authors. I believe this to limit the overall value of the work.
The other two weaknesses have been addressed to my satisfaction.
Overall, I maintain the "4" score.

**Limitations:**

yes

**Quality:**

3

**Strengths And Weaknesses:**

Strengths:

* 1 Addresses an important challenge facing transformers
* 2 Provides clear experimental promise of the proposed approach on algorithmic tasks
* 3 Provides diverse experimental results: algorithmic tasks, benefit from pretraining, mechanistic analysis
* 4 The paper is easy to read

Weaknesses:
* 1 The paper argues that length generalization transfer happens for “related” tasks (e.g., line 6), but it remains unclear what kind of relatedness here is needed. There is no theoretical (or even principled conceptual) understanding of how one would select a useful auxiliary task. The selection of auxiliary tasks is heuristic and exploratory. Hence, the paper provides no deeper insights for guiding the future design of auxiliary tasks. [UPDATE: the authors have some initial ideas based on MI, though no deeper principled insights are promised]
* 2 I found Figure 9 unconvincing. The caption claims that “All three metrics closely align” but other than maxima at 2000 and 16000 steps, I found little obvious alignment between the metrics. Also, (even if the alignment between the metrics were more obvious), the conclusion that “successful length generalization transfer accompanies shared attention mechanisms between tasks” is just based on correlational evidence [there is no causal evidence that attention map/ablation map similarity is causal to the generalization], and should be qualified accordingly. [UPDATE: the authors promise to address this]
* 3 In Figures 3 and 4, the training lengths vary (16+32 in Figures 3a,3b,4a,4c; 8+16 in Figure 3c, and similarly variable in Figures 5, 6). What is the motivation for this? Why not report unified results with the same lengths?  [UPDATE: the authors have explained the motivation]

---

> ### Author Rebuttal · Authors · 2025-07-31
>
> ### Thank you for your constructive review highlighting the clarity and importance of our work!
>
> Let us address each of the concerns below:
>
> $$ $$
> ### 1. Principled understanding of task "relatedness"
>
> In order to create a prescriptive formalism over the meaning of "relatedness", we are exploring the hypothesis that "tasks with high mutual information is more likely to have generalization transfer". Please see the details in reply to reviewer **8R8h**, as we dive deeper into this with some new experiment results.
> To thoroughly formalize and validate this hypothesis may be outside the scope of this work, in which we focus on elucidating the transfer phenomenon with tasks of different difficulty.
>
> $$ $$
> ### 2. Correlational evidence in figure 9
>
> We agree that the evidence in section 6.2 is only correlational and we will update the wording in the paper to reflect this.
>
> Furthermore, we will consider removing the "attention difference" metric since it serves the same purpose as "attention ablation map" and shows weaker correlation.
>
> Finally, we would like to point to figure 19, which supplements Fig 9 by showing the same analysis for more tasks in the addition family. Across these tasks, the local maxima, or "bumps" in the generalization gap (orange), mostly match with the maxima in the "head ablation map difference" (green).
>
> $$ $$
> ### 3. Motivation for different task lengths
> We choose different max lengths because the tasks have different character lengths for the same length parameter. For example, the multiplication task is much longer than the addition task, because it is solved using a chain of thought format. Thus we need to adjust the lengths we sample to make training feasible on our hardware. In Table 2, we provide examples of the task input/output format.
> Additionally in Sec 6.1, we show that the transfer phenomenon depends on the _relative_ lengths between the main and auxiliary tasks, rather than the absolute lengths we choose.
>
> $$ $$
> > In Figure 3c, it’s presupposed that when training Reverse Add up to length 8 on its own, it won’t length-generalize (which is not shown in Figure 3d). Is this the case?
>
> Yes, the lack of length generalization when training on single task is true for all lengths.
>
> >The paper shows transfer for related tasks, but not lack of transfer in the case of unrelated tasks. Have the authors tried to pair up unrelated or less-related tasks to show that indeed “related” tasks (line 6) are needed for transfer to happen?
>
> Section 4.4 shows that pairing reverse addition with the unrelated "copy-first" task yields no transfer, even though the auxiliary task provides longer inputs. This reinforces that structural relatedness, not mere exposure to long sequences, is necessary for transfer.

---

> ### Comment · Reviewer_BHPD · 2025-08-04
>
> Thanks to the authors for the response. Implementing these points will further improve the paper.
> The idea that high MI may be a measure of task relevance is interesting.

---

### Official Review · Reviewer_ySe2 · 2025-07-02

**Clarity:** 4
**Significance:** 3
**Originality:** 4
**Rating:** 5
**Confidence:** 3

**Summary:**

This paper investigates whether being trained on longer instances of an auxiliary task helps with length generalization on a main task. The authors find that it generally does, but only when the tasks are related and when the length difference is not too stark. They also find evidence that the improvement in generalization performance is associated with an increased sharing of mechanisms (as indicated by attention maps and ablation maps) between tasks.

**Questions:**

As noted above, I really liked this paper. My primary concern is the evidence in Section 6.2. I think a more extensive evaluation of the hypothesized association between mechanism sharing and length generalization would make the paper considerably stronger (whether or not this hypothesized association turns out to be true). In my comments above, I have provided one example for what such an analysis could look like. If this is not possible in the rebuttal period, I think it would be good to more strongly acknowledge the limitations of the current approach.

**Ethical Concerns:**

["NO or VERY MINOR ethics concerns only"]

**Final Justification:**

I continue to think this is a good paper that should be accepted to Neurips.

**Limitations:**

yes

**Paper Formatting Concerns:**

No concerns.

**Quality:**

3

**Strengths And Weaknesses:**

**Strengths**

I really liked this paper. The idea is clear and elegant, the results are convincing, and I appreciated that the authors studied (relatively small) pretrained language models as well. I also thought the paper was well written and I thought the schematic figures indicating the authors' hypotheses were very nice. The limitations section helpfully acknowledges questions that this paper does not address and highlights potential for future work. Finally, I thought it was good (and important) that the authors studied tasks from several domains.

**Weaknesses**

Primarily, I think Section 6.2 is currently lacking. It is both a little difficult to assess the extent to which ablation and attention difference are similar over time (other than them both going down) and there are many confounding factors, as many quantities vary which the training epoch (e.g. training set performance, in-distribution generalization, convergence to more consistent solutions from potentially more random initializations). I therefore think the evidence provided for more length-generalizing networks having more mechanisms in common is pretty weak. I think it would be interesting to compare the ablation and attention difference between different random seeds that exhibit different length generalization or between the networks exhibiting different length generalization in Figure 8. Finding an association between ablation/attention difference and length generalization across different networks would be much stronger evidence in my opinion.

As a more minor comment, I think it would be nice to connect the limitations section (which, as noted above, was generally good) to prior work that could be useful in addressing these questions.

---

> ### Author Rebuttal · Authors · 2025-07-31
>
> **Thank you for your encouraging review highlighting the clarity of our idea, results and presentation!**
>
> Let us address your concerns below:
>
> > It is both a little difficult to assess the extent to which ablation and attention difference are similar over time (other than them both going down) and there are many confounding factors, as many quantities vary which the training epoch (e.g. training set performance, in-distribution generalization, convergence to more consistent solutions from potentially more random initializations).
>
> We agree the analysis in sec 6.2 has much room to improve. In particular we will consider removing the "attention difference" plot since the correlation with generalization gap is weak. We'd also like to point out a few clarifying points:
>
> * In Fig 9 and 19, the training set performance and in distribution generalization saturates to 100% at step 2000, the first point on the x-axis. Therefore the plot shows entirely the progress of out-of-distribution performance.
> * The "bumps" of the OOD performance (generalization gap) is the result of unstable training dynamics (Sec A.1), this means throughout training, the model flip-flops between a generalizable solution and one that is not.
> * We show that whenever the generalization gap is high, so is the head ablation map difference (and vice versa), therefore linking the macroscopic metric with the mechanistic metric.
>
> > I think it would be interesting to compare the ablation and attention difference between different random seeds that exhibit different length generalization or between the networks exhibiting different length generalization in Figure 8.
>
> Yes, we agree. Figure 9 and 19 only shows the comparison for different tasks at different checkpoints, and we are working on incorporating further variations between different seeds.
>
> We will update the manuscript to acknowledge the limitations of the current mechanistic analysis.

---

> > ### Comment · Reviewer_ySe2 · 2025-08-01
> >
> > Thank you for your response; I appreciate your commitment to updating the manuscript to acknowledge the limitations of the current mechanistic analysis.

---

### Official Review · Reviewer_hBvs · 2025-07-03

**Clarity:** 4
**Significance:** 3
**Originality:** 4
**Rating:** 6
**Confidence:** 5

**Summary:**

This paper uncovers a novel finding adding to our understanding of transformer models: length generalisation can transfer across similar tasks in a variety of data, task, and model settings. They validate this on several task families, show instances of lack of generalisation due to task dissimilarity, show the finding holds on pretrained models, and provide interpretability analysis showing cross-task transfer is related to shared underlying mechanisms.

**Questions:**

- How were these tasks chosen? For the synthetic tasks, is it possible to formalise why task transfer is happening? (E.g. is the [C-RASP](https://arxiv.org/abs/2506.16055) program for these tasks sharing some key operations?) This isn't necessary for the paper as it stands, but I'm curious if there is some intuition on the authors part as to when generalisation does or does not occur across tasks.
- Can you explain figure 9 more? Particularly the leftmost subplot doesn't seem to be showing a strong trend. Maybe some error bars could help guage the significance of the correlation in the three metrics.

**Ethical Concerns:**

["NO or VERY MINOR ethics concerns only"]

**Final Justification:**

I am satisfied by the author responses and hope to see this paper accepted; it is a valuable contribution to the field.

**Limitations:**

yes

**Quality:**

4

**Strengths And Weaknesses:**

Strengths
- This is a really interesting and novel finding that adds to our understanding of language models. The overall idea is well-motivated and the paper is very nicely written, and the analyses (generalisation gap, different checkpoints, control task transfer) is comprehensive. Overall, I am very convinced by the narrative presented by the paper.
- I appreciate the inclusion of details of mechanistic results in sec. 6.2 and the relevant appendices, which elucidate why this happens.
- Further experiments comparing NoPE and RoPE are great to see; while these are in the appendix, this is important (along with the control task failures) to show that the finding is not merely about better learning of how to use RoPE but something more fundamental about cross-task transfer.

Weaknesses
- None apparent.

---

> ### Author Rebuttal · Authors · 2025-07-31
>
> **Thank you for your encouraging review pointing out the novel findings and detailed experiments!**
>
> Let us address your questions below
>
> > How were these tasks chosen?
>
> We picked the string and arithmetic tasks to be common tasks used in the length generalization literature [1] [2], and to increase the task complexity, we picked the maze solving task.
>
> > For the synthetic tasks, is it possible to formalise why task transfer is happening? (E.g. is the C-RASP program for these tasks sharing some key operations?)
>
> Formalizing the transfer condition is indeed an interesting future direction. Our current hypothesis is that task pairs that has higher mutual information is more likely transferrable, as measured by the mutual information of the task function applied to the input random variable. We provide more details in the rebuttal for reviewer 8R8h.
> We also believe that the tasks needs to share some internal mechanism in order for the transfer to happen, and the C-RASP framework as you have mentioned would definitely help us.
>
> > Can you explain figure 9 more? Particularly the leftmost subplot doesn't seem to be showing a strong trend.
>
> You are right to point out weak correlation of the "attention difference" metric comparing to other two metrics. Since its function is the same as the "head ablation map difference" metric, we will consider removing this metric.
>
> Looking at Fig 9 and Fig 19 together, the observation is that
> * For task pairs with successful transfer, the "head ablation map difference" metric correlates with generalization gap.
> * For task pairs without transfer, the same metric does not correlate with generalization gap.
>
> > Maybe some error bars could help guage the significance of the correlation in the three metrics.
>
> We display the metrics for individual seeds as opposed to aggregate over multiple seeds (with error bars) because we want to show that the local maxima or "bumps" in the generalization gap is mirrored on the other two metrics. Fig 19 shows more examples of this correlation in different tasks.
>
>
> $$ $$
> $$ $$
> $$ $$
> _References_
>
> [2] H. Zhou, A. Bradley, E. Littwin, N. Razin, O. Saremi, J. Susskind, S. Bengio, and P. Nakkiran,
> “What algorithms can transformers learn? A study in length generalization,”
> arXiv preprint arXiv:2310.16028, 2023.
>
> [3] E. McLeish, A. Bansal, A. Stein, N. Jain, J. Kirchenbauer, B. R. Bartoldson, B. Kailkhura, A. Bhatele, J. Geiping, A. Schwarzschild, et al., “Transformers can do arithmetic with the right embeddings,”
> arXiv preprint arXiv:2405.17399, 2024.

---

### Official Review · Reviewer_8R8h · 2025-07-04

**Clarity:** 3
**Significance:** 3
**Originality:** 4
**Rating:** 3
**Confidence:** 4

**Summary:**

This paper investigates a novel mechanism for improving length generalization in Transformers, termed "extrapolation by association." The central hypothesis is that a model can learn to generalize to longer, unseen input sequences for a "main task" if it is jointly trained on a related "auxiliary task" that includes examples of those longer sequences. The authors provide empirical evidence for this phenomenon, which they call "length generalization transfer," across a diverse set of algorithmic domains: arithmetic (e.g., reverse add), string manipulation (e.g., capitalize-reverse), and maze navigation (e.g., DFS trace).

The key findings are:
1. Co-training with a longer, related auxiliary task consistently enables the main task to extrapolate beyond its training length, a capability it lacks when trained in isolation.
2. This transfer is contingent on the structural similarity between tasks, as control experiments with unrelated tasks show no such benefit.
3. The effect is also observed when fine-tuning pretrained language models (SmolLM), and the capacity for generalization improves with the extent of pretraining, suggesting that pretraining builds a reusable "computational scaffolding."
4. Initial mechanistic analysis suggests that successful transfer correlates with the reuse of the same attention heads for both tasks, indicating a sharing of internal computational circuits.

**Questions:**

1. The instability of transfer due to random initialization (Sec A.1) is a significant concern, as it suggests the phenomenon might occur by chance rather than being a reliable property. Could you analyze the properties of the model initializations that lead to successful transfer versus those that fail? For instance, are there discernible statistical differences in initial weight norms or other properties? A positive result here would greatly increase my confidence in the robustness of your findings and could change my score.
2. The concept of "task relatedness" is central to your claims but is defined intuitively. Your post-hoc analysis showing a correlation with circuit sharing is insightful, but it doesn't help in predicting which tasks will work together beforehand. Have you considered any a priori metrics for quantifying task relatedness (e.g., based on algorithmic complexity, information theory, or embedding similarity of task descriptions) that might predict the likelihood or effectiveness of length generalization transfer?

**Ethical Concerns:**

["NO or VERY MINOR ethics concerns only"]

**Limitations:**

yes

**Quality:**

3

**Strengths And Weaknesses:**

Strength:
- The paper introduces a novel and significant research direction for length generalization, shifting focus from purely architectural solutions to the training paradigm itself.
- The experimental validation is thorough and well-designed, using three distinct task domains which strongly supports the generality of the core claim.
- The inclusion of control experiments with unrelated tasks provides compelling evidence that the observed transfer is due to task relatedness and not merely exposure to longer sequences.
- The investigation extends to pretrained models, bridging the gap between findings on synthetic tasks and their potential relevance to large language models.
- The mechanistic analysis using attention head ablation maps provides valuable, albeit initial, insight into how the transfer might be occurring at the circuit level.

Weakness:
- The primary weakness is the reported instability of the training dynamics; the success of the transfer is highly sensitive to random model initialization, which may limit the reliability and practical applicability of the phenomenon.
- The crucial concept of "task relatedness" is defined intuitively and only analyzed post-hoc; the work lacks a predictive, a priori metric to determine which tasks will facilitate transfer.
- The experiments are confined to relatively simple, deterministic algorithmic tasks, and it remains unclear how these findings would translate to more complex, ambiguous, or knowledge-intensive domains like natural language.

---

> ### Author Rebuttal · Authors · 2025-07-31
>
> **Thank you for your constructive review pointing out the novelty and thoroughness of our research!**
>
> Let us address the three main concerns below
>
> ## 1. Instability of the training dynamics
>
> To further investigate the source of training instability, we conducted two experiments that either 1) varied 5 seeds for model initialization keep dataset the same, or 2) varied 5 dataset seeds and keep the model initialization the same. However, in both cases, the transfer phenomenon is unreliable. This suggest that both model initialization and dataset composition contributes to the unstable generalization transfer. A previous work [5] similarly observed that "length generalization is not robust to neither weight initialization nor training data order".
>
> The third source of instability is from the training progress itself. Section A.1 shows that the generalization performance in fact oscillates in the same training run. Another work [1] also makes similar observations for hierarchical generalization, describing it as "competition" between two solutions (one generalizes, one does not). We will update these reference and discussions in the paper.
>
> Importantly though, the unstable training dynamics is only limited to the train-from-scratch case. Section A.5 repeat the same experiments for pretrained models and we observe successful length generalization across multiple seeds, suggesting more robustness for pretrained models.
>
> The transfer is also robust for highly correlated tasks. For example, training the following tasks together always results in robust transfer to longer addition.
> * task 1: short reverse addition with random padding on either side, e.g. 00012300+00045600
> * task 2: short reverse addition, e.g. 123+456
>
> Overall, We found that finetuning from pretrained model or having highly aligned tasks can result in robustness of such transfer.  However, it is unclear in the general case whether the instability is initialization-driven or data-driven, and it is an interesting future direction to find conditions for robust generalization transfer.
>
> $$ $$
> ## 2. A predictive, a priori metric for "task relatedness"
>
> This is an important direction, and we are working on one promising hypothesis, that "pairs of tasks with high mutual information will result in length generalization transfer". Concretely, assume we have two functions over random variables, $f$ and $g$, which takes the same random variable $Z$ to $X=f(Z)$ and $Y=g(Z)$, then define the estimated mutual information $$I(f;g)=I(X; Y) = \sum_{x \in \mathcal{X}} \sum_{y \in \mathcal{Y}} p(x, y) \log \left( \frac{p(x, y)}{p(x)\,p(y)} \right)$$
> We can then measure the mutual information between the task pairs that exhibit transfer and tasks that don't. For example, to estimate the mutual information for the addition task,
>
> * The input $Z$ are two integers sampled uniformly sampled over length $l$.
> * The functions of interest are
>     * reverse addition ($f$),
>     * carry only ($g$),
>     * no carry ($d$)
>     * reverse subtraction ($e$)
>     * Repeat first (control task) ($h$)
> * Suppose we want to estimate $I(f;g)$. We can sample a dataset $D_Z \sim p(Z)$, and apply our functions to the dataset $D_X=f(D_Z)$ and $D_Y=g(D_Z)$.
> * Finally we estimate the discrete mutual information using $D_X$ and $D_Y$ according to the formula.
>
> Estimating using task length 8 and 1 million examples, the results below shows that the control task has much less mutual information than the other tasks.
> - $I(f; g) = 1.7294$
> - $I(f; d) = 5.5450$
> - $I(f; e) = 0.7094$
> - $I(f; h) = 0.0331$ - Control task
>
> We believe this is a promising direction for formalizing the "task relatedness" concept, and we will include this framework in the paper if it passes more careful validation.
>
> We have also included the script for the example mutual information estimation at the end.
>
> $$ $$
> ## 3. More complex domains like natural language
>
> It is indeed a very interesting future direction to find a natural language setting to investigate length generalization transfer. For example, length generalization of math proof, or more complex synthetic settings like GSM-Infinite [4]. The challenge is that natural language tasks are almost always an amalgamation of many sub tasks, which makes it hard to isolate the transfer effect. We will update the limitations & future works section to include these specific examples.
>
> In this work, we mainly follow and expand upon the experimental settings in the line of algorithmic length generalization literature [2] [3], focusing on smaller-scale, controlled experiments.
>
> We hope these clarifications address your concerns and will gladly provide further details in the camera‑ready if accepted.
>
> $$ $$
> $$ $$
> $$ $$
> $$ $$
> $$ $$
>
> _References_
>
> [1] T. Qin, N. Saphra, and D. Alvarez-Melis, Sometimes I am a Tree: Data Drives Unstable Hierarchical Generalization, arXiv:2412.04619, 2024
>
> [2] H. Zhou, A. Bradley, E. Littwin, N. Razin, O. Saremi, J. Susskind, S. Bengio, and P. Nakkiran,
> “What algorithms can transformers learn? A study in length generalization,”
> arXiv preprint arXiv:2310.16028, 2023.
>
> [3] E. McLeish, A. Bansal, A. Stein, N. Jain, J. Kirchenbauer, B. R. Bartoldson, B. Kailkhura, A. Bhatele, J. Geiping, A. Schwarzschild, et al., “Transformers can do arithmetic with the right embeddings,”
> arXiv preprint arXiv:2405.17399, 2024.
>
> [4] Y. Zhou, H. Liu, Z. Chen, Y. Tian, and B. Chen, “GSM-Infinite: How Do Your LLMs Behave over Infinitely Increasing Context Length and Reasoning Complexity?,” arXiv preprint arXiv:2502.05252, 2025.
>
> [5] Y. Zhou, U. Alon, X. Chen, X. Wang, R. Agarwal, and D. Zhou, “Transformers Can Achieve Length Generalization But Not Robustly,” arXiv preprint arXiv:2402.09371, Feb. 2024.
>
> _MI Estimation Script_
> ```python
> import numpy as np
> from sklearn.metrics import mutual_info_score
> from collections import Counter
> import pandas as pd
> import seaborn as sns
> import matplotlib.pyplot as plt
>
> def int_to_bits(n, length):
>     return [int(b) for b in np.binary_repr(n, width=length)][::-1]
>
> def reverse_add(a_bits, b_bits):
>     carry = 0
>     result = []
>     for a, b in zip(a_bits, b_bits):
>         total = a + b + carry
>         result.append(total % 2)
>         carry = total // 2
>     return result
>
> def carry_bits(a_bits, b_bits):
>     carry = 0
>     result = []
>     for a, b in zip(a_bits, b_bits):
>         total = a + b + carry
>         result.append(int(total >= 2))
>         carry = total // 2
>     return result
>
> def repeat_first(a_bits, b_bits):
>     return a_bits + a_bits
>
> def digitwise_sum_mod_10(a_bits, b_bits):
>     return [(a + b) % 10 for a, b in zip(a_bits, b_bits)]
>
> def reverse_subtract(a_bits, b_bits):
>     borrow = 0
>     result = []
>     for a, b in zip(a_bits, b_bits):
>         diff = a - b - borrow
>         if diff < 0:
>             diff += 2
>             borrow = 1
>         else:
>             borrow = 0
>         result.append(diff)
>     return result
>
> def bits_to_tuple(bits):
>     return tuple(bits)
>
> def generate_samples(n_samples, bit_length):
>     max_val = 2 ** bit_length
>     a_vals = np.random.randint(0, max_val, size=n_samples)
>     b_vals = np.random.randint(0, max_val, size=n_samples)
>     return a_vals, b_vals
>
> def compute_mutual_info(f_outs, g_outs):
>     unique_f = list(set(f_outs))
>     unique_g = list(set(g_outs))
>     f_map = {x: i for i, x in enumerate(unique_f)}
>     g_map = {x: i for i, x in enumerate(unique_g)}
>     f_encoded = [f_map[x] for x in f_outs]
>     g_encoded = [g_map[y] for y in g_outs]
>     return mutual_info_score(f_encoded, g_encoded)
>
> def joint_distribution(x_outs, y_outs):
>     joint_counts = Counter(zip(x_outs, y_outs))
>     df = pd.DataFrame([
>         {'x': x, 'y': y, 'count': count}
>         for (x, y), count in joint_counts.items()
>     ])
>     return df
>
> def plot_joint_distribution(df, title):
>     pivot = df.pivot_table(index='x', columns='y', values='count', fill_value=0)
>     plt.figure(figsize=(10, 8))
>     sns.heatmap(pivot, cmap="Blues", cbar_kws={'label': 'Frequency'})
>     plt.title(title)
>     plt.xlabel('Output of second function')
>     plt.ylabel('Output of f')
>     plt.tight_layout()
>     plt.show()
>
> def main(l=8, n_samples=1000000):
>     a_vals, b_vals = generate_samples(n_samples, l)
>
>     f_outs, g_outs, h_outs, d_outs, e_outs = [], [], [], [], []
>
>     for a, b in zip(a_vals, b_vals):
>         a_bits = int_to_bits(a, l)
>         b_bits = int_to_bits(b, l)
>         f_outs.append(bits_to_tuple(reverse_add(a_bits, b_bits)))
>         g_outs.append(bits_to_tuple(carry_bits(a_bits, b_bits)))
>         h_outs.append(bits_to_tuple(repeat_first(a_bits, b_bits)))
>         d_outs.append(bits_to_tuple(digitwise_sum_mod_10(a_bits, b_bits)))
>         e_outs.append(bits_to_tuple(reverse_subtract(a_bits, b_bits)))
>
>     mi_results = {
>         "I(f; g)": compute_mutual_info(f_outs, g_outs),
>         "I(f; h)": compute_mutual_info(f_outs, h_outs),
>         "I(f; d)": compute_mutual_info(f_outs, d_outs),
>         "I(f; e)": compute_mutual_info(f_outs, e_outs),
>     }
>
>     print("Mutual Information Results:")
>     for k, v in mi_results.items():
>         print(f"{k} = {v:.4f}")
>
>     # Plotting
>     # plot_joint_distribution(joint_distribution(f_outs, g_outs), "Joint Distribution of f and g")
>     # plot_joint_distribution(joint_distribution(f_outs, h_outs), "Joint Distribution of f and h")
>     # plot_joint_distribution(joint_distribution(f_outs, d_outs), "Joint Distribution of f and d")
>     # plot_joint_distribution(joint_distribution(f_outs, e_outs), "Joint Distribution of f and e")
>
> if __name__ == "__main__":
>     main()
> ```

---

> > ### Author Response · Authors · 2025-08-08
> > **Follow-up on Rebuttal**
> >
> > Dear Reviewers and AC,
> >
> > Thanks for the thoughtful reviews. In our rebuttal we addressed the main concerns (1) instability of training dynamics, (2) a predictive, a priori metric for "task relatedness" and (3) more complex domains like natural language. If anything remains unclear, we’re happy to provide further detail. If these points resolve the concerns you noted, we’d appreciate an updated assessment.

---

### Note · Authors · 2025-08-13

We sincerely thank all reviewers and the AC for their engagement. Across the reviews, there is broad agreement that our work introduces length generalization transfer as a **novel and significant research direction**, validated through **diverse experiments** spanning arithmetic, string, and maze tasks, with both from-scratch and pretrained models. Reviewers highlighted the **clarity of writing** and **comprehensive evaluation**.

We acknowledge and addressed each limitations raised during discussion:
- Instability in the train-from-scratch setting. We added experiments to show caused by initialization and data composition. However, we also showed robustness when finetuning from pretrained models or using highly aligned task pairs.
- Task relatedness currently defined post-hoc; we proposed and demonstrated a **mutual information–based metric** as a concrete step toward a priori selection.
- Mechanistic analysis is correlational. So we committed to clarifying scope, removing weaker metrics, and expanding seed/task comparisons in future work.
- Extension to complex natural language domains remains an open direction. We emphasized that our setting inherits and expands upon the experimental settings of other past works studying length generalization. We also suggested controlled synthetic–natural hybrids (e.g., GSM-Infinite) as a potential new setting to bridge this gap.

One reviewer (8R8h) did not respond despite AC follow-up; we trust our rebuttal addressed their concerns. We believe the paper’s novelty, breadth of evidence, and clear future path make it a meaningful contribution to understanding and leveraging length generalization transfer.

---

### Decision · Program_Chairs · 2025-09-17

**Decision:**

Accept (spotlight)

**Comment:**

This paper proposes a mechanism for length generalization in transformers. It works by co-training for length generalization using a sufficiently similar task, such that extrapolation on the original task can take place via association.

Reviewers are (very) positive about this work, highlighting the novelty of the approach relative to prior works that focus on architectures for addressing the length generalization problem, and the overall quality of the paper. Some weaknesses, such as the mechanistic analysis and quantifying the notion of task-relatedness, were only partially addressed in the rebuttal, but are not in the way of publication as is also evident from the reviewer scores. One reviewer, reviewer 8R8h who is recommending weak reject, did not take part in the author-reviewer discussion even after repeated attempts from the AC to contact them. Overall, I agree with the other reviewers that this is a valuable contribution, potentially even worth highlighting at the conference.